

# Point Clouds to Primitives
## Tool for reconstructing geometric primitives from point clouds and semantic labelling



**Autors**: Michał Klemens ⦿ · Jakub Mikłasz ⦿ · Aleksander Musz ⦿ · Maja Placek ⦿

**Supervisor:** Marcin Jodłowiec

### Abstract

The project concentrates on the reconstruction of 3D primitive shapes from a point cloud. The tool is an application that identifies basic shapes such as spheres, cones, cylinders, or planes, and then visualises them in a 3D space. It also enables the user to semantically label generated results by providing classes, properties, and hierarchising them. Object semantics created in the process of labelling are visualised on diagrams. The solution is of significant technical relevance, supporting the automation of geometric analysis and the creation of high-quality training datasets for applications such as optimising 3D models in manufacturing or training methods to detect objects in point clouds.

## 1 INTRODUCTION

The project addresses the problem of automatic analysis and simplified reconstruction of objects from spatial data in the form of a point cloud. Using advanced 3D model estimation algorithms, the tool identifies basic shapes such as spheres, cones, cylinders or planes, and then visualises them in a 3D space.

The user has the option of assigning semantic labels (names, classes, properties) to the detected shapes, which enables accurate labelling of model elements. For example, in a rabbit model, the application can detect two cylinders representing ears, and the user will assign them the label 'rabbit ears'. This kind of process makes it possible to create structures that describe the semantics of the modelled objects, which can be used to train artificial intelligence systems.

The aim of the project is to simplify the process of preparing geometric data for applications such as optimising 3D models in manufacturing or training algorithms to detect objects in point clouds. The solution is of significant technical relevance, supporting the automation of geometric analysis and the creation of high-quality training data.

## 2 RELATED WORK

### 2.1 *Random Sample Consensus*

The *Random Sample Consensus* algorithm (*RANSAC*) [1] originally developed for image analysis and automated mapping is commonly used for shape detection in 3D point clouds both as a stand-alone method and as a component of the [4, 7, 11] detection process. It involves iterative random selection of minimal sets of points that define a particular geometric primitive. Once the primitives are created, they are tested against all points in the data to determine how many of them represent the shape score. The shape with the best score in a given iteration is added to the result.

The main advantages of the *RANSAC* algorithm include:

· Simple implementation and easy extensibility of the algorithm

· High noise resistance

· Versatility of applications in various fields

## 2.2  *Point Cloud Library*

*Point Cloud Library* (*PCL*) described in detail in the paper [6], played a crucial role in the implementation of the project. It is an open-source library designed for point cloud processing and analysis. It offers a wide range of tools, including filtering, segmentation, registration and simplification functions for 3D data. The library supports multiple data formats and integrates with visualisation systems such as *VTK*. *PCL* also includes an implementation of model estimation algorithms, such as *RANSAC*, which enables easy detection of basic geometric shapes in point clouds. Applications of *PCL* include robotics, spatial mapping, 3D object analysis and data visualisation.

## 2.3  Similiar product – *Cloud Compare*

*CloudCompare* is a popular tool for point cloud visualisation and analysis. *RANSAC Shape Detection* plug-in *CloudCompareRANSAC* [8] enables the detection of basic 3D shape primitives, such as planes, spheres, cylinders or cones, in point clouds. The operation of the plug-in is based on the *RANSAC* algorithm.

Point Cloud to Primitives application is distinguished by the functionality of semantic labelling of found models. Given the ability to assign semantics to the detected shapes, the user can give context to the analysed objects. Not only does this facilitate data interpretation and presentation, but it also enables the preparation of high-quality training datasets for artificial intelligence systems.

# 3  TECHNOLOGY STACK

## 3.1  *C++*

*C++* [10] was chosen as the main programming language for the application because of its effectiveness, low-level memory control, and broad support for 3D data processing libraries, such as *Point Cloud Library* (*PCL*).

## 3.2  *Qt* i *QML*

*Qt Framework* in combination with *QML* [9] language enabled the creation of modern, interactive user interface. In particular *Qt Quick 3D* was used to visualise point clouds and detected shapes in the 3D scene.

## 3.3  *Point Cloud Library* (*PCL*)

*PCL* offers a wide range of tools for processing and analysis of point clouds, among others, *RANSAC* algorithm implementation. The library provides solutions for the modification of detection parameters (e.g. error tolerance, model distance threshold) and integration of additional, user-provided analysis methods. *PCL* is able to efficiently manage the memory and parallel processing of data, reducing the processing time of the algorithm.

## 3.4  *MinGW*

At the beginning *MSVC* compiler was chosen because of its compatibility with precompiled libraries for *Windows* operating system (mainly *PCL* which is available as compiled binary files). However, later it was found out that *MSVC* compiler lacks full support for parallel processing with *OpenMP* (which *PCL* uses) which lead to the lack of the ability to fully parallelise the detection process. Due to the fact that it is advisable to use parallelism (especially for large point clouds), the decision was made to switch to *MinGW* compiler [5]. This change required compiling *PCL* and its dependencies from source.

## 3.5  *D2*

*D2* (*Declarative Diagramming*) [3] is a scripting, diagram language that produces diagrams from text. *D2* provides an API in *Golang* language which enables creating and editing diagrams programmatically. This project uses it to automatically generate diagrams from semantic descriptions created by the user.

# 4 RESULTS

## 4.1 Architecture

The application consists of several specified components, which are responsible for: loading the point cloud (*PointCloudHandler*), detection of 3D primitives (*ShapeDetecion*), management of user-defined classes (*ClassSystem*) and objects (*ObjectSystem*) as well as the user interface (*User interface*). The user interface has been implemented in the QML language, using *Qt Quick* visual components along with models and 3D scene from *Qt Quick 3D*. Other components have been implemented using C++ and *Qt Framework*. Point cloud loading and shape detection use normal estimation and segmentation algorithms from *PCL*, respectively. The application is an executable file with dependencies (*Qt, PCL*) included in the form of dynamically linked libraries.

Diagram generator (*DiagramGenerator*) is a separate executable file, written in the *Golang* programming language, and run by the program as a separate process.

The application architecture is illustrated in a component diagram (Fig. 1).

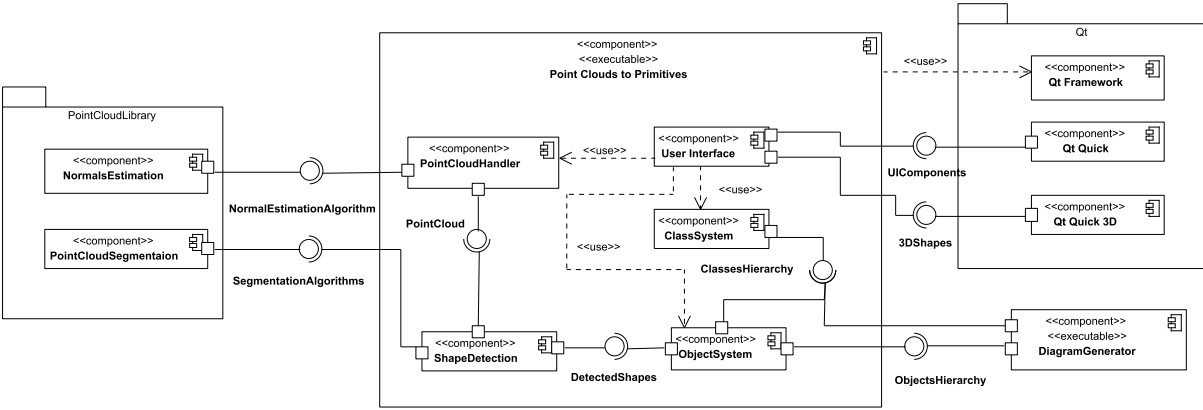

Figure 1: Component diagram showing the application architecture

## 4.2 Functionality

- The user can create a new project based on a point cloud file.

- The loaded point cloud is processed (point normals are estimated) and displayed inside the 3D scene.

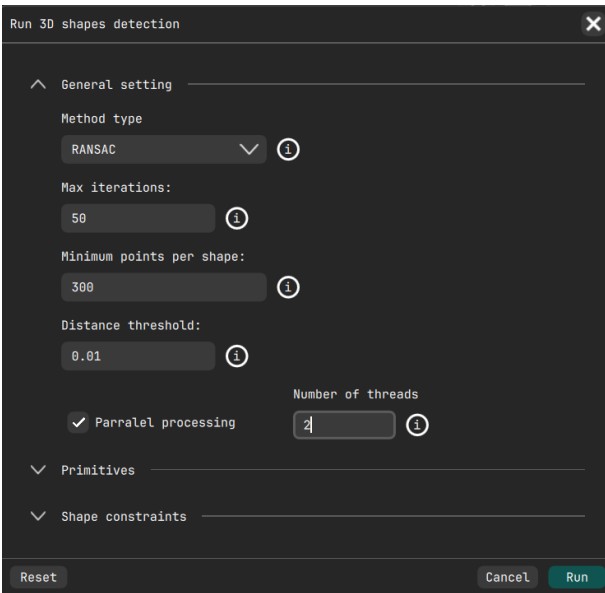

Figure 2: Example configuration of shape detection parameters.

- Movement inside the 3D scene is done using the mouse and keyboard, with the option of switching to rotation around the object.

- The user can configure detection parameters, such as: distance threshold, the types of shape being detected, the algorithm used, minimum number of points making up each shape, as well as the option to run detection in multiple threads, and other advanced settings. (Fig. 2)

- Detected shapes are visualized within the 3D scene, with the option of toggling the visibility of the point cloud and the 3D models.

- In the tree view users can edit the shape hierarchy, group objects, edit names, and add semantic meaning to elements.

- Users can define new objects' classes, relationships between objects and assign various properties to them.

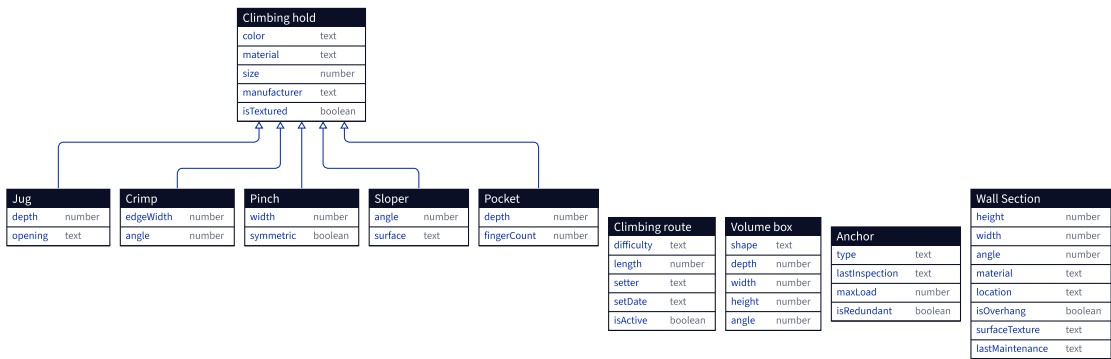

Figure 3: Example of a generated class diagram

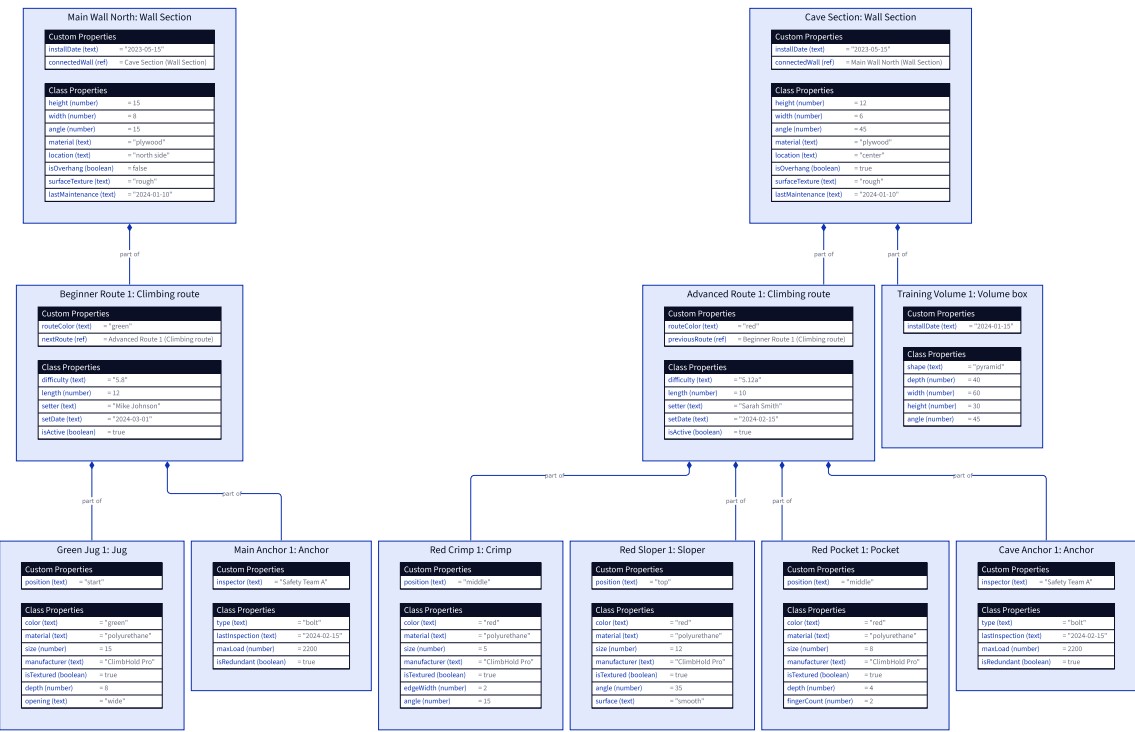

Figure 4: Example of a generated object diagram

- Class diagrams (Fig. 3) and object diagrams can be generated (Fig. 4) from a semantically described model.

## 4.3 Shape detection

In the performed tests, the tool based on the *PCL* library correctly identified primitive geometric shapes such as spheres, cylinders, cones and planes, in the analysed point cloud. Model estimation algorithms, in particular *RANSAC* have shown high accuracy when processing point clouds of simpler 3D models, in which individual elements can be easily represented using basic shapes, as shown in the example (Fig. 5). However, for complex objects that do not contain elements which can be easily mapped using the indicated three-dimensional primitives, the results have been less satisfactory. In these situations, the algorithm proved to be less accurate in fitting the models to the point cloud, indicating the need to modify or extend the processing methods available in the *PCL* library, to better meet the needs of the project in future iterations.

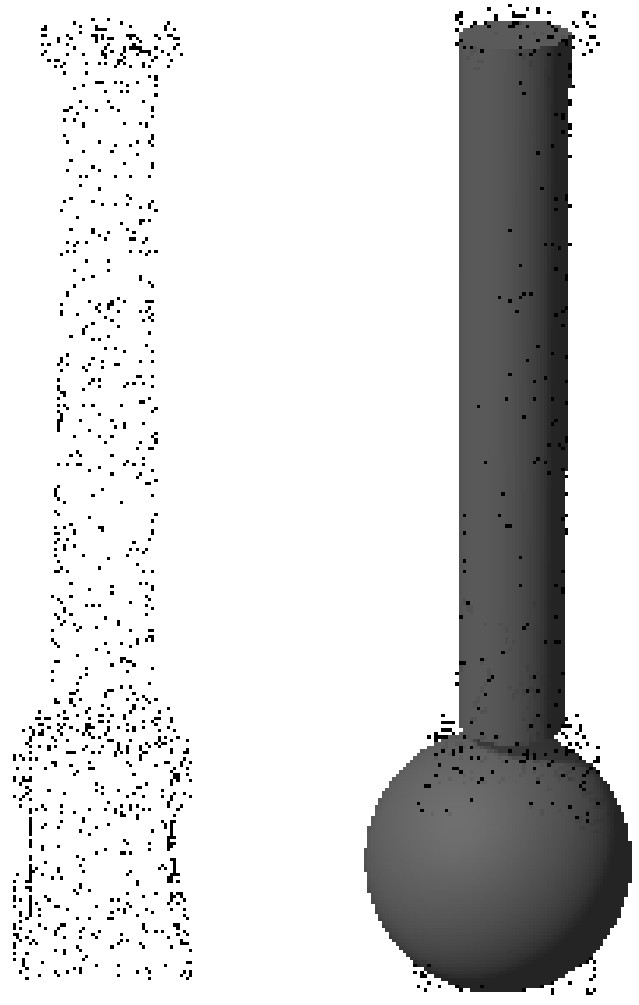

Figure 5: Example of a processed point cloud [2]

## 4.4 Semantic description

Semantic description is based on user-defined classes (which can be assigned to objects) and properties (including object name) along with objects' hierarchy. Classes are named containers with properties, enriched with an inheritance and subtyping mechanism, while each property consists of a name and value type. Supported types are: character string (*text*), real number (*numerical*) and reference to an object (*reference*). The class inherits properties from all its base classes in the hierarchy. Moreover,

the subtyping mechanism upgrades the resulting model by providing additional information – class-instance relationships between all base classes for the class assigned to an object and that object. When assigning a class to an object, a user is required to provide values for all properties this class contains. To simplify the resulting model, every class is constrained to have at most one base class, and every object to have at most one class assigned. However, the user can assign additional properties and their values to individual objects. Every object also has a special text property – name.

The second component of the semantic description is defining a hierarchy of objects in the form of a tree using grouping and *drag and drop* mechanism. The class and properties can be assigned to every detected object and to every group.

These components altogether enable the description of any object in a systematic and easy-to-process way, which is ideal for labelling data for use in training of artificial intelligence methods. Moreover, properties and class hierarchy, along with objects' properties (with values) and their hierarchy, are visualised in the form of diagrams (exemplary diagrams in Fig. 3 and Fig. 4, respectively). For better readability, visualisation of reference type properties' values as connections on the diagram was not included in the project.

# 5 CONCLUSIONS

The project's primary contribution is a comprehensive platform that seamlessly integrates geometric processing and semantic modelling capabilities. This integration enables full control over the complete 3D data pipeline, encompassing detection, visualisation, semantic organisation, and analysis. The application combines advanced functionality with intuitiveness, making it a versatile solution for a wide range of applications.

One of the most important strengths of this tool is its ability to optimise and compress 3D data. The algorithms used achieve a significant reduction in geometric complexity while maintaining key geometric features and semantic properties. This efficient data management is particularly valuable for resource-demanding applications, such as real-time 3D simulations and *VR/AR* environments, where rendering performance directly impacts user experience.

The resulting semantic networks can be used to build a knowledge base for use in machine learning. This makes it possible not only to classify objects but also to develop more advanced analysis and prediction systems that can be used in fields such as robotics, intelligent manufacturing systems, and autonomous vehicles.

By integrating all of these functions into a single tool, the project provides an all-encompassing solution that simplifies, optimises, and enhances the process of working with 3D data while opening up new possibilities in modelling, analysis, and machine learning.

# 6 FUTURE DIRECTIONS

## 6.1 Optimization of reconstruction algorithms

A potential direction for the project could be to expand the functions offered by *PCL* library or to create a stand-alone library, better suited to the needs of the platform. In the course of the work, many problems were encountered due to the limitations of existing algorithms, such as the difficulty in accurately reconstructing complex shapes. *PCL* offers a very limited set of primitive shapes, adding shapes such as a cuboid, a torus, etc. could prove valuable to the detection process. Furthermore, a desirable functionality could be a model constructed with the *CSG* (*Constructive Solid Geometry*) method. In this way, it would be possible to recognise a cylinder with a hollow sphere, etc. Current solutions in *PCL* often require additional optimisation or manual adjustment of parameters, significantly prolonging the process. The development of dedicated algorithms could improve reconstruction accuracy, increase efficiency, and facilitate integration with data analysis tools. Further development in this direction could also contribute to a more versatile tool for other users involved in point cloud processing.

## 6.2 Training artificial intelligence models on the basis of semantic networks

A way forward for the project could also be to use semantic networks as a basis for training artificial intelligence models. The data generated in the labelling process could serve to create a knowledge base to facilitate automatic recognition and classification of objects in point clouds. This approach could contribute to the development of more advanced AI models capable of analysing complex scenes and understanding the relations between objects. Furthermore, the knowledge base could support other projects that require a semantic understanding of a 3D space, enabling their development without the

need for manual data preparation. As a result, this solution could improve the automation and precision of geometry reconstruction and analysis processes in various applications.

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
