# OpenReview forum: "Point Clouds to Primitives - Tool for reconstructing geometric primitives from point clouds and semantic labelling"
_pwr.edu.pl/Wrocław_University_of_Science_and_Technology/2024/ZPI_Day — Wrocław University of Science and Technology 2024 ZPI Day Submission_

### Official Review · Reviewer_rZfA · 2024-12-06
**The review of Point Clouds to Primitives**

**Confidence:** 5
**Significance Of Results:** 4
**Overall Quality:** 5

**Compliance With Template:**

5: Very High Quality – The article contains all the required sections, which are written in a very detailed, clear, and error-free manner. The structure is professional and meets expectations, and the content adheres to the highest substantive and formal standards.

**Description Of Results:**

4: High Quality – The results are described in detail and supported by usage examples or evaluations. The description is reliable but may lack full depth of analysis.

**Feedback On Consistency:**

The paper is clear, easy to read, thus well-written. It contains all the sections needed for technical description of such works. Some quantification of expected and achieved results would be beneficial.

**Potential For Development:**

The results achieved and maturity of elaborated methods is moderate, however it has strong potential for future development. In terms of front-end, the app is ready to present results for future algorithms and novelties.

**Project Nature Evaluation:**

Authors have described in the projects their method and tools that they have used.

**Technical Language Precision:**

5: Very High Quality – The language is entirely appropriate for a technical report. All terms are used correctly and precisely, and the style is professional, clear, and coherent, without any errors or ambiguities.

---

### Official Review · Reviewer_ZnEv · 2024-12-06
**The article presents a well-structured and innovative approach, offering practical solutions with clear potential for future development and application.**

**Confidence:** 4
**Significance Of Results:** 4
**Overall Quality:** 5

**Compliance With Template:**

5: Very High Quality – The article contains all the required sections, which are written in a very detailed, clear, and error-free manner. The structure is professional and meets expectations, and the content adheres to the highest substantive and formal standards.

**Description Of Results:**

5: Very High Quality – The results are described in detail, clearly and comprehensively, supported by thorough evaluation, analysis, and convincing usage examples. The description meets the highest substantive standards.

**Feedback On Consistency:**

The project description is consistent and well-organized, with a logical progression from problem analysis to the presentation of results and conclusions. Each section aligns seamlessly, effectively building upon the previous one to provide a clear and coherent narrative.

**Potential For Development:**

The article clearly outlines potential directions for future development, providing well-defined and practical possibilities for expanding the project. The suggestions, such as optimizing reconstruction algorithms and leveraging semantic networks for AI training, are thoughtfully presented and demonstrate a strong understanding of the field. These directions are detailed and offer valuable insights into how the project could evolve to address current limitations and enhance its utility.

**Project Nature Evaluation:**

The project exhibits clear characteristics of engineering work, demonstrating a high level of utility through practical applications and problem-solving. It effectively applies advanced technical methods and innovative technological solutions, showcasing a structured and methodical approach to addressing complex challenges.

**Technical Language Precision:**

5: Very High Quality – The language is entirely appropriate for a technical report. All terms are used correctly and precisely, and the style is professional, clear, and coherent, without any errors or ambiguities.

---

### Official Review · Reviewer_ESek · 2024-12-07
**Point Clouds to Primitives - Tool for reconstructing geometric primitives from point clouds and semantic labelling**

**Confidence:** 2
**Significance Of Results:** 4
**Overall Quality:** 4

**Compliance With Template:**

4: High Quality – The article contains all the required sections, which are well-written and substantively correct, although minor errors or shortcomings may be present. The overall structure is clear and coherent.

**Description Of Results:**

4: High Quality – The results are described in detail and supported by usage examples or evaluations. The description is reliable but may lack full depth of analysis.

**Feedback On Consistency:**

- The conclusions are consistent with the results but could more explicitly connect back to the original problem statement.
- The recommendations are practical, but the rationale behind them needs further explanation to ensure clarity.
Suggestions: Offer actionable recommendations (e.g., restructuring sections, adding examples, or improving data presentation).

**Potential For Development:**

The project exhibits strong potential for development but would benefit from a more detailed exploration of practical applications and a roadmap for future work.

**Project Nature Evaluation:**

Overall, the project aligns well with engineering principles but could be improved by including a prototype or practical validation of the proposed solutions.

**Technical Language Precision:**

4: High Quality – The language is appropriate for a technical report. Terminology is used correctly, and statements are precise, with only minor shortcomings that do not affect the overall clarity.

---

### Decision · Program_Chairs · 2024-12-10

Accept (Oral)